# Attitudes and current practice in alcohol screening, brief intervention, and referral for treatment among staff working in urgent and emergency settings: An open, cross-sectional international survey

Holly Blake[1,2]*, Mehmet Yildirim[1], Vinishaa Premakumar[3], Lucy Morris[4], Philip Miller[5], Frank Coffey[4]

1 School of Health Sciences, University of Nottingham, Nottingham, United Kingdom, 2 NIHR Nottingham Biomedical Research Centre, Nottingham, United Kingdom, 3 School of Medicine, University of Nottingham, Nottingham, United Kingdom, 4 Emergency Department, Nottingham University Hospitals NHS Trust, Nottingham, United Kingdom, 5 East Midlands Academic Health Sciences Network, Nottingham, United Kingdom

* holly.blake@nottingham.ac.uk

## Abstract

### Background

The aim of the study was to ascertain the views and experiences of those working in urgent and emergency care (UEC) settings towards screening, brief intervention, and referral to treatment (SBIRT) for alcohol, to inform future practice.

### Objectives

To explore i) views towards health promotion, ii) views towards and practice of SBIRT, iii) facilitators and barriers to delivering SBIRT, iv) training needs to support future SBIRT practice, and v) comparisons in views and attitudes between demographic characteristics, geographical regions, setting and occupational groups.

### Methods

This was an open cross-sectional international survey, using an online self-administered questionnaire with closed and open-ended responses. Participants were ≥18 years of age, from any occupational group, working in urgent and emergency care (UEC) settings in any country or region.

### Results

There were 362 respondents (aged 21–65 years, 87.8% shift workers) from 7 occupational groups including physicians (48.6%), nurses (22.4%) and advanced clinical practitioners (18.5%). Most believed that health promotion is part of their role, and that SBIRT for alcohol prevention is needed and appropriate in UEC settings. SBIRT was seen to be acceptable to

**Data Availability Statement:** The data underlying the results presented in the study are available

from The Nottingham Research Data Management Repository (doi: 10.17639/nott.7292).

**Funding:** "The study was funded by Nottingham University Hospitals Charity (Ref: APP 2346/FR-000000340). The funder provided support in the form of contributions to salaries for authors [LM, HB], but did not have any additional role in the study design, data collection and analysis, decision to publish, or preparation of the manuscript. The specific roles of these authors are articulated in the 'author contributions' section".

**Competing interests:** The authors have declared that no competing interests exist.

patients. 66% currently provide brief alcohol advice, but fewer screen for alcohol problems or make alcohol-related referrals. The most common barriers were high workload and lack of funding for prevention, lack of knowledge and training on SBIRT, lack of access to high-quality resources, lack of timely referral pathways, and concerns about patient resistance to advice. Some views and attitudes varied according to demographic characteristics, occupation, setting or region.

## Conclusions

UEC workers are willing to engage in SBIRT for alcohol prevention but there are challenges to implementation in UEC environments and concerns about workload impacts on already-burdened staff, particularly in the context of global workforce shortages. UEC workers advocate for clear guidelines and policies, increased staff capacity and/or dedicated health promotion teams onsite, SBIRT education/training/resources, appropriate physical spaces for SBIRT conversations and improved alcohol referral pathways to better funded services. Implementation of SBIRT could contribute to improving population health and reducing service demand, but it requires significant and sustained commitment of time and resources for prevention across healthcare organisations.

## Introduction

It is globally accepted that health promotion is an effective tool for improving population health. The World Health Organization (WHO) has long advocated the value of promoting health through settings, such as hospitals, schools, prisons, workplaces, communities, villages, and cities [1]. The WHO Health Promoting Hospitals movement with its roots in the Ottawa Charter, encouraged hospitals to place greater emphasis on health promotion and disease prevention, rather than on diagnostic and curative services alone. In England, the National Health Service encourages staff to use everyday interactions with patients to discuss healthy lifestyle changes as part of 'Making Every Contact Count' (MECC) [2], which is seen to be a valuable approach to improving population health. This is based on the premise that the volume of citizens that encounter healthcare services daily provides a wealth of contacts between clinicians and patients that provide potentially valuable 'teachable moments' for discussions about health behaviour change [3].

Screening, Brief Intervention, and Referral to Treatment (SBIRT) is a public health approach to identifying alcohol users, providing brief advice, and referring them on to rehabilitation and recovery services as appropriate. Studies have shown brief interventions delivered in emergency departments (EDs) to be effective and potentially cost-effective [4–6]. EDs are increasingly being identified as an important setting for capitalising on 'teachable moments' that can be used to reduce health-comprising behaviours (e.g., hazardous alcohol consumption, injury prevention, risky driving, cigarette smoking, poor diet, lack of exercise, and sleep deficit) and reduce demand on the healthcare system [7–9]. Nonetheless, research and practice in this area remains limited and EDs have not systematically engaged with prevention. An umbrella review of systematic reviews and meta-analyses shows that urgent and emergency care (UEC) settings are both under-researched and under-utilised for delivering health promotion activities, although alcohol prevention is gathering more traction [10].

An evidence synthesis located very few studies investigating the barriers to implementing health promotion interventions among emergency care workers–most of the research was conducted in the US, Canada, or Australia. Only ten surveys were identified (only one of which included data from the UK, Scotland only) and facilitators to health promotion were very poorly captured [11]. Studies highlighted mixed views over whether health promotion should be part of the emergency personnel job role. There was some discomfort in broaching lifestyle behaviour conversations, with many workers highlighting a lack of competence or training in health promotion. The review called for more research to establish whether incorporating health promotion into the roles of staff in UEC settings is acceptable [11].

The aim of this study was to ascertain the views and experiences of those working in urgent and emergency care (UEC) settings towards screening, brief intervention, and referral (SBIRT) for alcohol in order to inform future health promotion practice. To achieve this, the objectives were to quantitatively explore i) views towards health promotion in UEC settings, ii) views towards and practice of SBIRT, iii) facilitators and barriers to delivering SBIRT in UEC settings, iv) training needs to support future SBIRT practice, and v) comparisons in views and attitudes between demographic characteristics (age, gender, ethnicity), geographical region (UK/international), type of UEC setting (ED/other) and occupational groups (physician, registered nurse, ACP or other). Survey free-text responses allowed qualitative exploration of the key issues associated with the delivery of SBIRT in UEC from the perspective of UEC staff.

## Methods

### Study design and setting

This was an open cross-sectional international survey, using an online self-administered questionnaire with closed and open-ended responses. Reporting was guided by the Strengthening the Reporting of Observational Studies in Epidemiology (STROBE) statement [12] (S1 Table) and the Checklist for Reporting Results of Internet E-Surveys (CHERRIES) [13]. The setting was UEC (such as emergency departments, trauma units, urgent care centres, minor injury units, walk-in centres, pre-hospital / ambulance services).

### Study population

Individuals ≥18 years of age, from any occupational group, working in UEC settings in any country or region, were eligible to participate. Participants were excluded if they were <18, or did not work in an UEC setting.

### Sampling and sample size

We adopted a convenience sampling strategy, targeting a sample size of 300–400 for a maximum sampling error of 5.0–5.8% [14]. The survey was open to anyone who worked in UEC services at the time of the study.

### Study procedures

This was an open survey developed and tested in February 2022, hosted on a secure, web-based platform (Jisc Online Surveys). For context, the survey took place two years after coronavirus (COVID-19) was declared a pandemic by the World Health Organization in March 2020, following sustained (and ongoing) clinical pressures, workforce shortages and burnout in the health and care workforce. Several different distribution channels and collection methods were used in effort to increase reach across occupational groups and settings and minimise sampling bias. An invitation (S1 File) containing a link to the study information and online

survey was widely distributed via email and social media (Twitter, Facebook, LinkedIn) to UEC professional networks and special interest groups, educational mailing lists at higher education and healthcare institutions, and regular mailings and publications. Data were collected from March to December 2022. The participant information sheet included the purpose of the study and specified that the survey would take approximately 10 minutes to complete. Participants were informed what electronic data were stored, where, and for how long, and that data were held securely, password protected and only accessible to the research team, for whom contact details were provided. Data were captured automatically from those who accessed and completed the survey online. Study promotion was intentionally broad to include diverse geographic regions. Participation was voluntary and anonymous to minimise social desirability bias. No incentives were offered for survey completion. In effort to minimise non-response bias, the purpose and goals of the study were clearly outlined in the participant information sheet, and reminders were posted at least weekly via different distribution channels (e.g., emails, social media).

## Data collection

The questionnaire survey was drafted by the study team, which included researchers with expertise in public health and health behaviour change (HB, MY), emergency nurses (PM, LM) and an emergency physician (FC). Items were developed based on expert opinion, published literature and a prior exploratory survey in a local emergency department. Survey items were finalised through public involvement consultation with two nurses who were not part of the study team. The survey was pilot tested (paper and online version) with health researchers, nurses and advanced clinical practitioners working in acute care settings (n = 8) to check appropriateness of content, usability, and technical functionality. The survey was used to collect quantitative data including responses on a Likert scale and categorical responses with non-response options (i.e., 'not applicable', 'prefer not to say'), and qualitative data in free text responses. There were 30 questions divided into four sections over 29 pages (S2 File):

i. sociodemographic (10 questions, all multiple choice, of which 4 included free text options). Sociodemographic characteristics focused on age, gender, ethnicity, highest qualification, occupational group, length of service, employment setting and geographical region, and work pattern.

ii. views towards health promotion in UEC settings (6 questions, all multiple choice, of which 1 included free text option).

iii. views towards and practice of SBIRT (8 questions, of which 4 included free text options and 1 contained 15 Likert scale sub-items).

iv. facilitators and barriers to SBIRT, including training needs (6 questions, including 6 Likert scale, 3 multiple choice and 3 free text options).

Items were not randomised or alternated. Items were visible one section at a time and were conditionally displayed based on responses to other items, to reduce the number and complexity of the questions. Participants were able to review and change their answers through a 'Back' button. Survey completion was anonymous, although participants were given the option of providing an email address to be contacted for future research and informed that this would be separated from their survey responses if provided.

## Data analysis

The participation rate is the ratio of those who participated divided by the number of first survey page visitors. The completion rate is the ratio of the number of people who finished the

survey divided by those who completed the first page of the survey [14]. No surveys were excluded from analysis; *n* varied according to number of completers per item. Quantitative data were analysed using STATA SE, version 17.0 [15]. Comparative analysis assessed differences in views and practices relating to SBIRT according to demographic characteristics (age; gender: male/female; ethnicity: White/others), geographical region (UK/non-UK), type of UEC setting (ED/other) and occupational group (physician/registered nurse/ACP/other). Gender categories 'non-binary/gender fluid' and 'prefer not to say' were excluded from comparisons due to small sample size. Independent categorical variables were compared using Chi-square test or Fisher's exact test. Statistical significance between groups was determined by comparing z-scores obtained from adjusted residuals and new p-values adjusted using the Bonferroni method. Other comparisons involved non-parametric Mann-Whitney U Tests (two groups) and non-parametric Kruskal-Wallis Tests (more than two groups). Dunn's test was conducted to identify which specific characteristics contributed to significant differences. Qualitative responses from free text items were analysed using thematic analysis, conducted by two researchers (MY and VP) using the software NVivo (released in March 2020) [16]. Data were coded independently, and the researchers established main themes based on their codes and groupings. Themes were then discussed between three researchers (MY, VP and HB) and synthesised into a single report. Any discrepancies and disagreements were resolved through discussion between all three researchers.

## Ethical considerations

This study was performed in accordance with the Declaration of Helsinki and all relevant guidelines and regulations. The study researchers were trained in Good Clinical Practice. Information about the aims and conduct of the study were provided. Data were treated in confidence and analysed anonymously. Participants were informed that by voluntarily completing and submitting the online survey they were providing their written consent to take part. This survey study was the first part of a wider programme of research on alcohol prevention in urgent and emergency care, for which the protocol was approved by the University of Nottingham Faculty of Medicine and Health Sciences Research Ethics Committee on 19 January 2022 (FMHS 415–1121), with an extension to the study end date approved on 28 September 2022.

## Results

### Study population

A total of 362 people completed the survey. The participation rate was 12.7% (362 completed the survey of 2836 views) and the completion rate was 100%. Sample characteristics are shown in Table 1. Participants were aged between 21 and 65 years (female 57.7%, male 41.2%, non-binary/gender fluid 0.6%, preferred not to disclose 0.6%). They had worked in their current role <1 year to >35 years (66.7% between 1–10 years). Respondents worked in the United Kingdom (n = 307, 84.8%: England n = 265 (86.3%), Wales n = 3 (1.0%), Scotland n = 33 (10.7%), Northern Ireland n = 6 (2.0%)), Europe (n = 23, 6.3%), North America/Central America (n = 11, 3.0%), Africa (n = 8, 2.2%), Asia (n = 6, 1.6%), Australia/New Zealand (n = 7, 2.0%). Most worked in a hospital emergency department (n = 310, 85.6%).

There were respondents from seven occupational groups: physician, registered nurse (RN), advanced clinical practitioner (ACP; advanced role, not limited to traditional boundaries of clinical specialisms), paramedic, nurse practitioner (NP), nursing/healthcare assistant (N/HCA), and linked professions. Most of the respondents were physicians, nurses, or advanced clinical practitioners, and worked shifts.

**Table 1. Descriptive characteristics of participants.**

| Characteristics | Whole sample (N = 362) | UK (N = 307, 84.8%) | International (N = 55, 15.2%) |
|---|---|---|---|
| **Age[a] (n = 362)** | | | |
| 21–30 | 66 (18.2) | 54 (17.6) | 12 (21.9) |
| 31–40 | 108 (29.9) | 89 (29.0) | 19 (34.5) |
| 41–50 | 136 (37.6) | 117 (38.1) | 19 (34.5) |
| 51–65 | 52 (14.4) | 47 (15.3) | 5 (9.1) |
| **Gender (n = 362)** | | | |
| Male | 149 (41.2) | 123 (40.1) | 26 (47.2) |
| Female | 209 (57.7) | 180 (58.7) | 29 (52.8) |
| Non-binary/Gender fluid | 2 (0.6) | 2 (0.6) | 0 |
| Prefer not to disclose | 2 (0.6) | 2 (0.6) | 0 |
| **Ethnicity (n = 356)** | | | |
| White—British | 230 (63.5) | 227 (74) | 3 (5.6) |
| White—Irish | 25 (6.9) | 10 (3.3) | 15 (27.2) |
| White—Other White background | 28 (7.7) | 16 (5.2) | 12 (21.9) |
| Mixed–White and Black Caribbean | 3 (0.8) | 3 (1.0) | 0 (0) |
| Mixed–White and Black African | 2 (0.6) | 0 (0) | 2 (3.6) |
| Mixed–White and Asian | 5 (1.4) | 4 (1.3) | 1 (1.8) |
| Mixed–Other Mixed background | 3 (0.8) | 1 (0.3) | 2 (3.6) |
| Asian/Asian British–Indian | 26 (7.2) | 20 (6.5) | 6 (11) |
| Asian/Asian British–Pakistani | 13 (3.6) | 10 (3.3) | 3 (5.6) |
| Asian/Asian British–Bangladeshi | 4 (1.2) | 2 (0.6) | 2 (3.6) |
| Asian/Asian British–Other Asian background | 7 (1.9) | 4 (1.3) | 3 (5.6) |
| Black/Black British–Caribbean | 0 (0) | 0 (0) | 0 (0) |
| Black/Black British–African | 10 (2.8) | 4 (1.3) | 6 (11) |
| Black/Black British–Other Black background | 0 (0) | 0 (0) | 0 (0) |
| Chinese | 4 (1.2) | 4 (1.3) | 0 (0) |
| Chinese—Other ethnic background | 2 (0.6) | 2 (0.6) | 0 (0) |
| **Highest Qualification (n = 362)** | | | |
| Degree level or above | 344 (95.0) | 291 (94.8) | 53 (96.3) |
| Another kind of qualification | 17 (4.7) | 15 (4.9) | 2 (3.7) |
| No qualifications | 1 (0.3) | 1 (0.3) | 0 (0) |
| **Occupational Group (n = 362)** | | | |
| Physician | 176 (48.6) | 146 (47.5) | 30 (54.5) |
| Registered Nurse (RN) | 81 (22.4) | 64 (20.8) | 17 (31.0) |
| Advanced Clinical Practitioner (ACP) | 67 (18.5) | 65 (21.2) | 2 (3.6) |
| Paramedic | 14 (3.9) | 10 (3.3) | 4 (7.3) |
| Nurse Practitioner (NP) | 10 (2.8) | 10 (3.3) | 0 (0) |
| Nursing/Healthcare Assistant (N/HCA) | 7 (1.9) | 7 (2.3) | 0 (0) |
| Linked profession (Linked SC)[b] | 7 (1.9) | 5 (1.6) | 2 (3.6) |
| **Employment setting (n = 362)** | | | |
| Emergency Department | 310 (85.6) | 259 (84.3) | 51 (92.8) |
| Urgent Care Centre | 17 (4.7) | 15 (4.9) | 2 (3.6) |
| Pre-hospital/ambulance | 16 (4.5) | 15 (4.9) | 1 (1.8) |
| Acute Medicine | 14 (3.8) | 14 (4.5) | 0 (0) |
| Minor Injury Unit | 5 (1.4) | 4 (1.3) | 1 (1.8) |
| **Target populations (n = 362)** | | | |
| Adults (> = 18 years) | 121 (33.4) | 103 (33.5) | 18 (32.7) |

(*Continued*)

**Table 1.** (Continued)

| Characteristics | Whole sample (N = 362) | UK (N = 307, 84.8%) | International (N = 55, 15.2%) |
|---|---|---|---|
| Children and Young People (<18 years) | 8 (2.2) | 8 (2.6) | 0 (0) |
| Both | 233 (64.4) | 196 (63.9) | 37 (67.3) |
| **Work pattern (n = 362)** | | | |
| Shifts | 318 (87.8) | 268 (87.3) | 50 (90.9) |
| Standard office hours (e.g., 9–5) | 44 (12.2) | 39 (12.7) | 5 (9.1) |

[a]There were no respondents in the 16–20 years category. [b]Linked profession: linked to / working directly with to ED patients, such as ED-based social prescribing; drug and alcohol liaison team; addiction psychiatrist; acute/critical care medicine physician.

### Views towards health promotion in UEC settings

Of respondents, 361 (99.7%) viewed health promotion to be moderately or very important, and 315 (87.0%) believed that health promotion was part of their role, 29 (8.0%) were unsure. There were no significant differences in views towards the importance of health promotion according to demographic characteristics, geographical region, type of UEC setting or occupational group. However, nurses were significantly more likely than other occupations to agree that health promotion was part of their role ($X^2$(6, N = 362) = 12.66, p = .040). One hundred and one respondents (27.9%) reported that they had undertaken training related to brief interventions for lifestyle behaviours. This training primarily consisted of minimal health promotion content delivered within taught university courses (for many, this was not in recent years), or informal learning through meetings, webinar and conference attendance. A few respondents, primarily advanced practitioners, had undertaken e-learning on MECC, and training in motivational interviewing techniques. Almost three quarters of respondents had not undertaken any training in this area (n = 261, 72.1%), most of whom were shift workers (n = 236, 74.2%).

Respondents indicated that, in addition to alcohol prevention, there is a role for promoting diverse areas of health in UEC settings. These areas include physical activity (n = 242, 66.9%), diet / nutrition (n = 249, 68.8%), weight / body mass index (n = 246, 68.0%), health screening (n = 243, 67.1%), vaccination uptake (n = 226, 62.4%) or other (n = 53, 14.6%).

Other areas of prevention that respondents felt could be incorporated into UEC settings included: drug misuse, smoking cessation, medication/treatment adherence, sexual health, violence (youth violence, domestic violence), gambling, healthcare utilisation, sleep hygiene, self-care, and basic first aid. There was variation in views about whether there is a role in UEC for other types of health promotion. Compared to female workers, male workers reported a higher perceived level of difficulty in promoting physical activity (z = -2.46, p = .012), diet (z = -3.01, p = .002), weight (z = -2.37, p = .012), or engaging in health screening (z = -2.55, p = .013) in UEC. Workers from ethnic minorities reported less perceived difficulty with promoting diet (z = -2.45, p = .016), and weight (z = -2.06, p = .034), and engaging in health screening (z = -2.41, p = .014) in UEC compared to White workers. Workers in ED settings reported a higher perceived level of difficulty in promoting diet (z = -2.18, p = .021) and weight (z = -2.59, p = .009) compared to workers based in other types of UEC setting. Physicians were less likely to believe there is a role for health screening in UEC settings compared to nurses or ACPs ($X^2$(3, N = 362) = 13.56, p = .003). Finally, workers from the UK perceived a higher level of difficulty implementing health screening (z = -2.92, p = .003) and vaccination (z = -2.56, p = .019) in UEC compared to workers outside of the UK.

## Views towards and practice of SBIRT

**Views towards screening, brief advice, and referral to treatment (SBIRT) for alcohol.**
Most respondents (n = 333, 92.0%) agreed that there is a need for SBIRT for alcohol in UEC
settings The majority believed that UEC is an appropriate place to deliver SBIRT for alcohol
with regards to screening (n = 324, 89.5%), brief advice (n = 333, 92.0%) or referrals (n = 310,
85.6%). In terms of the practicalities of delivery, views varied. Of respondents, 35.6% (n = 139)
believed that it is practical to implement SBIRT for alcohol in UEC, although 49.7% (n = 180)
were uncertain or felt it depended on the circumstances. There were no significant differences
in views towards the need for SBIRT, or its practicality in UEC according to demographic
characteristics, geographical region, type of UEC setting or occupational group. Professionals
working outside of the UK were more likely to express a belief that screening ($X^2$(1, N = 362) =
5.19, p = .024) and referral ($X^2$(1, N = 362) = 8.29, p = .004) are appropriate for implementa-
tion in UEC, compared to professionals working in the UK. However, it is worth noting that
the majority of respondents (within and outside of the UK) reported that SBIRT is appropriate
to implement in UEC.

Respondents' level of agreement with statements relating to SBIRT are shown in Table 2.
Here, percent represents the number and proportion of respondents within each occupation
that agreed or strongly agreed with each statement. Younger workers (aged 21–30) were less
likely to report knowing how to screen for alcohol consumption ($X^2$(3, N = 362) = 11.32, p =
.018) knowing how to make alcohol-related referrals ($X^2$(3, N = 362) = 18.87, p < .001), and
having the skills ($X^2$(3, N = 362) = 12.78, p = .005) and confidence ($X^2$(3, N = 362) = 14.84, p =
.002) to make referrals, compared to older workers (aged 41–50). Female workers were more
likely than male workers to report that conversations about alcohol with patients was sup-
ported in their organisation ($X^2$(1, N = 358) = 7.53, p = .006). Workers from ethnic minority
groups were more likely to agree that delivering SBIRT will ultimately decrease UEC atten-
dance and hospitalisations, compared to White workers ($X^2$(1, N = 362) = 5.40, p = .027).
Respondents working outside of the UK were more likely to agree with this statement than
those working in the UK ($X^2$(1, N = 362) = 8.15, p = .004). Workers from the UK were more
likely (than those working outside the UK) to report knowing how to screen patients for alco-
hol consumption ($X^2$(1, N = 362) = 7.78, p = .005), being confident in their ability to screen
($X^2$(1, N = 362) = 6.98, p = .008) and having access to the resources and information they
needed to discuss alcohol with patients ($X^2$(1, N = 362) = 7.51, p = .006).

Physicians were more likely than workers from other occupations to agree that they know
how to screen for alcohol consumption ($X^2$(3, N = 362) = 25.53, p < .001) have the skills ($X^2$(3,
N = 362) = 36.29, p < .001) and confidence ($X^2$(3, N = 362) = 27.65, p < .001) to conduct
screening. Physicians were also more likely than workers from other occupations to report
having the skills ($X^2$(3, N = 362) = 26.82, p < .001) and confidence ($X^2$(3, N = 362) = 14.97, p =
.002) to have brief conversations with patients about reducing alcohol consumption, and to
report knowing how to make alcohol referrals ($X^2$(3, N = 362) = 13.88, p = .003), and having
existing skills ($X^2$(3, N = 362) = 12.23, p = .007) and confidence ($X^2$(3, N = 362) = 14.53, p =
.002) to make referrals. There were no significant differences in the level of agreement with
any of the statements according to the type of UEC setting that respondents worked in.

**Experience of SBIRT in practice.** Of respondents, 234 (64.6%) had delivered some ele-
ments of SBIRT for alcohol in the past. Respondents aged 21–30 years were significantly less
likely to report SBIRT experience compared to other age groups ($X^2$(3, N = 362) = 9.45, p =
.022). Regarding occupational groups, physicians reported greater SBIRT delivery experience,
while nurses displayed comparatively less experience when compared to other professions
($X^2$(3, N = 362) = 17.33, p = .001). No other significant differences were found with

**Table 2.  Agreement with statements relating to screening, brief advice, and referrals (SBIRT) by occupational group.**

| Statement (% agree or strongly agree) | Total N = 362 (%) | Physician n = 176 | RN n = 81 | ACP n = 67 | Paramedic n = 14 | NP n = 10 | N/HCA n = 7 | Linked n = 7 |
|---|---|---|---|---|---|---|---|---|
| **Delivering SBIRT will ultimately decrease urgent and emergency care attendance and hospitalisations** | 250 (69.0) | 121 (68.7) | 59 (72.8) | 43 (64.2) | 11 (78.6) | 6 (60.0) | 5 (71.4) | 5 (71.4) |
| **Reinforcing advice about alcohol use to patients in ED will prompt them to seek help** | 285 (78.7) | 136 (77.3) | 68 (84.0) | 54 (80.6) | 12 (85.7) | 6 (60.0) | 4 (57.1) | 5 (71.4) |
| **Having conversations about alcohol with my patients is supported in my organisation** | 297 (82.0) | 143 (81.2) | 67 (82.7) | 57 (85) | 10 (71.4) | 8 (80.0) | 6 (85.7) | 6 (85.7) |
| **Having conversations about alcohol with my patients is supported by my colleagues** | 300 (82.8) | 150 (85.2) | 65 (80.2) | 53 (79.1) | 10 (71.4) | 10 (100.0) | 5 (71.4) | 7 (100.0) |
| **My colleagues have conversations with their patients about alcohol** | 264 (72.9) | 133 (75.6) | 54 (66.7) | 47 (70.1) | 9 (64.3) | 9 (90.0) | 6 (85.7) | 6 (85.7) |
| **I understand the concept of a brief intervention for alcohol prevention** | 314 (86.7) | 152 (86.3) | 73 (90.1) | 57 (85.0) | 11 (78.5) | 8 (80.0) | 6 (85.7) | 7 (100.0) |
| **I know how to screen patients for alcohol consumption** | 249 (68.7) | 139 (79.0) | 47 (58.0) | 47 (70.1) | 3 (21.4) | 5 (50.0) | 3 (42.8) | 5 (71.4) |
| **I know how to give brief advice to patients about reducing alcohol consumption** | 275 (75.9) | 137 (77.8) | 56 (69.1) | 54 (80.6) | 11 (78.5) | 7 (70.0) | 5 (71.4) | 5 (71.4) |
| **I know how to make alcohol referrals** | 237 (65.4) | 129 (73.3) | 45 (55.5) | 45 (67.1) | 4 (28.5) | 5 (50.0) | 3 (42.8) | 6 (85.7) |
| **I have the skills to screen patients for alcohol consumption** | 249 (68.7) | 145 (82.4) | 42 (51.8) | 45 (67.1) | 6 (42.8) | 5 (50.0) | 1 (14.3) | 5 (71.4) |
| **I have the skills to give brief advice to patients about reducing alcohol consumption** | 257 (70.9) | 136 (77.3) | 46 (56.8) | 50 (74.6) | 10 (71.4) | 6 (60.0) | 4 (57.1) | 5 (71.4) |
| **I have the skills to make alcohol referrals** | 260 (71.8) | 143 (81.2) | 46 (56.8) | 52 (77.6) | 6 (42.8) | 5 (50.0) | 2 (28.6) | 6 (85.7) |
| **I am confident in my ability to screen patients for alcohol consumption** | 210 (58.0) | 123 (69.9) | 34 (42.0) | 40 (59.7) | 2 (14.3) | 4 (40.0) | 2 (28.6) | 5 (71.4) |
| **I am confident in my ability to have a conversation with patients about reducing alcohol consumption** | 258 (71.2) | 132 (75.0) | 44 (54.3) | 52 (77.6) | 11 (78.5) | 7 (70.0) | 6 (85.7) | 6 (85.7) |
| **I am confident in my ability to making alcohol referrals** | 223 (61.6) | 124 (70.4) | 41 (50.6) | 41 (61.2) | 5 (35.7) | 4 (40.0) | 2 (28.5) | 6 (85.7) |
| **I have access to the resources and information I need to discuss alcohol with patients** | 153 (42.2) | 83 (47.1) | 25 (30.8) | 29 (43.3) | 4 (28.5) | 3 (30.0) | 3 (42.8) | 6 (85.7) |

demographic characteristics. The elements of SBIRT that UEC workers had delivered in the past included alcohol screening (n = 151, 64.8%), brief advice (n = 219, 93.6%), and referrals (n = 176, 75.2%). In their experience of SBIRT, respondents viewed it to be acceptable to patients (screening: 95.3%, brief advice: 91.7%, referrals: 93.7%). The proportion of respondents currently engaging in SBIRT was lower. Less than half were currently screening patients for alcohol in their practice (n = 165, 45.6%), although two-thirds were providing brief advice (n = 240, 66.3%). Half of the sample was currently making alcohol-related referrals of any type (n = 183, 50.6%). Most indicated a willingness to deliver SBIRT for alcohol in the future (n = 323, 89.2%) and this did not differ according to demographic characteristics, geographical region, or type of UEC setting. Those who had ever screened for alcohol had used a range of tools. The most commonly used were Alcohol Use Disorders Identification Test (AUDIT) [17] (n = 52, 14.4%), Alcohol Use Disorders Identification Test for Primary Care (AUDIT-PC) [18] (n = 9, 2.5%), Alcohol Use Disorders Identification Test for Consumption (AUDIT-C) [19] (n = 80, 22.1%), Fast Alcohol Use Screening Test (FAST) [20] (n = 54, 14.9%), Paddington Alcohol Test (PAT) [21] (n = 49, 13.5%), Modified Single Alcohol Screening Questionnaire (M-SASQ) [22] (n = 11, 3.0%). Those selecting 'other' (n = 30, 8.3%) could not recall the tool name, had used non-standardised question items developed within their institution, or had

**Table 3. Current practices and experiences of alcohol referral.**

| Referral type (n = 362) | Current referral practice available in setting (n, % Yes) | Own referral experience (n, % Yes) |
|---|---|---|
| Psychological Treatment / Services (e.g., Psychotherapy, Cognitive Behavioural Therapy, Dialectical Behavioural Therapy, Behavioural Couples Therapy) | 80 (22.1) | 35 (9.7) |
| Specialist Alcohol Counselling | 217 (60.0) | 166 (45.9) |
| Brief Intervention (e.g., Motivational Interviewing, Solutions-Focused Approach) | 71 (19.6) | 47 (13.0) |
| 12-Step Facilitation Programme (e.g., Alcoholics Anonymous) | 42 (11.6) | 14 (3.9) |
| Inpatient unit or a medically supported residential service | 69 (19.0) | 55 (15.2) |
| Intensive community rehabilitation programme | 34 (9.3) | 17 (4.7) |
| Social network and environment-based therapies | 24 (6.6) | 13 (3.6) |
| Lifestyle intervention (e.g., yoga, meditation) | 15 (4.1) | 11 (3.0) |
| Creative therapy (e.g., art and music therapy) | 5 (1.4) | 3 (0.8) |
| Referral to GP/community drug and alcohol services | 33 (9.1) | 18 (5.0) |

used the CAGE Alcohol Abuse Screening Tool [23], Leeds Dependence Questionnaire (LDQ) [24], CIWA-AR Assessment for Alcohol Withdrawal [25], or the Glasgow Modified Alcohol Withdrawal Scale (GMAWS) [26]. One hundred and seventy-one (47.2%) respondents specified that they had never used any standardised tool.

Respondents who had delivered brief advice had done so by providing verbal education or advice (n = 310, 85.6%), brochure/leaflet (n = 107, 29.6%) or signposting to a website (n = 139, 38.4%). Any 'other' (n = 13, 3.6%) brief advice respondents referred to had been provided by another health professional or service, rather than themselves.

Current practices regarding referral to treatment with respect to alcohol, and participants' personal experiences of referral are reported in Table 3, respondents selected all that applied. Specialist alcohol counselling was most often identified as current referral practice in their setting. Fewer than one in five workers identified brief alcohol intervention as a current referral practice, with only 13.0% having referred a patient themselves for brief alcohol intervention themselves; referrals were more often to other types of intervention or service. For all referral types, there were very few respondents with personal experience of making the referral.

## Facilitators and barriers to delivering SBIRT in UEC settings

Key barriers to delivering SBIRT for alcohol are shown in Table 4, respondents selected all that applied.

All respondents identified multiple barriers to implementing SBIRT in UEC environments. There were no significant differences in reported barriers with age, gender, or type of UEC setting, although differences were observed with ethnicity, occupation, and geographical region. Workers from ethnic minority groups were more likely than White respondents to report barriers to SBIRT including: lack of knowledge on how to start a conversation about alcohol with a patient ($X^2$(1, N = 362) = 14.93, p < .001), lack of personal interest in prevention ($X^2$(1, N = 362) = 7.66, p = .006), lack of reimbursement for prevention ($X^2$(1, N = 362) = 29.31, p < .001), and expectation that patients would deny an alcohol issue ($X^2$(1, N = 362) = 6.23, p = .013). Ethnic minority workers were also more likely to report that their own alcohol consumption affects their willingness to engage in alcohol prevention than White workers ($X^2$(1,

**Table 4. Barriers to delivering SBIRT in UEC settings.**

| Barrier (N = 362) | Overall n (% Yes) | Identified as a barrier for: | | |
| --- | --- | --- | --- | --- |
| | | Screening n (%) | Brief Intervention n (%) | Referral n (%) |
| Lack of training | 297 (82.0) | 204 (80.3) | 224 (82.2) | 192 (75.6) |
| Lack of knowledge on which patients are suitable | 193 (53.3) | 140 (80.5) | 137 (78.8) | 143 (82.2) |
| Lack of knowledge on the process | 270 (74.6) | 164 (73.2) | 168 (75.0) | 193 (86.2) |
| Lack of knowledge on how to start a conversation with a patient | 160 (44.2) | 112 (84.2) | 120 (90.2) | 102 (76.7) |
| Lack of knowledge about the effectiveness | 232 (64.1) | 152 (77.6) | 179 (91.3) | 174 (88.8) |
| Not enough time / workload is too heavy | 319 (88.1) | 218 (85.5) | 233 (91.4) | 212 (83.1) |
| Lack of personal interest | 82 (22.7) | 80 (94.1) | 79 (92.9) | 74 (87.1) |
| Lack of reimbursement | 83 (22.9) | 70 (95.9) | 70 (95.9) | 69 (94.5) |
| Expected patient denial of alcohol issue | 175 (48.3) | 123 (90.4) | 105 (77.2) | 98 (72.1) |
| Expected patient resistance to advice | 224 (61.9) | 112 (65.9) | 148 (87.1) | 139 (81.8) |
| Lack of high-quality patient information and resources | 250 (69.1) | 139 (74.3) | 171 (91.4) | 167 (89.3) |
| Lack of clinical pathways | 248 (68.5) | 132 (72.5) | 143 (78.6) | 171 (94.0) |
| Own alcohol consumption affects willingness to engage in SBIRT | 32 (8.8) | 41 (97.6) | 39 (92.9) | 40 (95.2) |

N = 362) = 9.89, p = .002). With regards occupation, nurses were more likely than other occupations to report barriers including: lack of training ($X^2$(3, N = 362) = 15.85, p = .001), lack of knowledge about the process of SBIRT ($X^2$(3, N = 362) = 14.12, p = .003), lack of reimbursement for prevention ($X^2$(3, N = 362) = 11.69, p = .009), expected patient denial of an alcohol issue ($X^2$(3, N = 362) = 12.44, p = .006) and a lack of high-quality resources for prevention ($X^2$(3, N = 362) = 11.75, p = .008). Nurses, physicians and ACPs were more likely to report lack of time / heavy workload as a barrier to SBIRT compared with those from other occupations ($X^2$(3, N = 362) = 12.84, p = .005). Workers from outside the UK were more likely (compared to workers in the UK) to report certain barriers to SBIRT, including: lack of knowledge on which patients are suitable for SBIRT ($X^2$(1, N = 362) = 8.06, p = .005), the SBIRT process ($X^2$(1, N = 362) = 5.50, p = .015) and how to start a conversation with a patient about alcohol ($X^2$(1, N = 362) = 11.88, p = .001), and a lack of reimbursement for prevention ($X^2$(1, N = 362) = 15.73, p < .001).

**Training needs to support future SBIRT practice.** Respondents identified training needs which, if addressed, would facilitate SBIRT practice. Most respondents (n = 324, 89.5%) identified at least one training need. Training needs included medical complications of alcohol use (n = 96, 26.5%), social and psychiatric problems faced by people with alcohol use disorders (n = 115, 31.8%), screening and early identification of alcohol use disorders (n = 199, 55.0%), techniques for delivery of brief interventions (n = 242, 66.9%), treatment options for people with alcohol use disorders / problems (n = 232, 64.6%), diagnosing and treating alcohol withdrawal (n = 104, 28.7%), alcohol abstinence / reduction strategies (n = 165, 45.6%), counselling strategies to increase patients' motivation to cut down/abstain (n = 205, 56.6%), making referrals to relevant services (n = 194, 53.6%). Fourteen respondents (4.3%/324) provided additional free-text comment relating to training needs. Of these, 13 (4.0%/324) specified that all the listed areas were valuable and/or relevant, while just one respondent indicated that none of the areas listed were training needs.

## Qualitative findings

The latter part of the survey consisted of free text questions which invited additional comments from respondents regarding the delivery of SBIRT in UEC. Of respondents, 238/362 (65.7%) provided a comment. Two main themes were identified from the responses: (i) perceived barriers and enablers of SBIRT delivery in UEC and, (ii) role conflict.

**i. Perceived barriers and enablers of SBIRT delivery in UEC.** The vast majority of comments aligned with this theme. Respondents proposed a range of factors that hindered the delivery of SBIRT in UEC. Although UEC was described as "*an important setting for health promotion*" (ACP, Urgent Care Centre, Female) many referred to a lack of staff time for prevention activities, due to a long-standing workforce crisis within the healthcare systems with staffing shortages and heavy workloads exacerbated by the COVID-19 pandemic: "*it's very busy with COVID and long delays in emergency*" (Nurse, ED, Female); "*Not enough workforce to manage demand expectation on the number of patients we see every hour*" (ACP, Urgent Care Centre, Male); "*The service is functioning vastly outside of its capacity with staff barely able to deliver minimum expected services*" (Physician, ED, Male). Some expressed concerns that the implementation of SBIRT would place additional strain on staff who are already dealing with heavy workloads in UECs and were struggling for time to provide the basics of emergency care: "*Currently too busy to take this [SBIRT] on*" (Physician, ED, Male); "*with ongoing mismatch of capacity and demand, it is a struggle to provide good basic care, let alone gold standard*" (Physician, ED, Female). Respondents commonly felt ill-equipped to deliver SBIRT due to a lack of education and training relating to prevention, reporting "*very little knowledge about health promotion and prevention*" (Nurse, ED, Female) and a "*lack of understanding of MECC and why it is important in urgent care settings*" (Physician, Urgent Care Centre, Male). Others perceived that the physical environment was inappropriate to hold conversations with patients about alcohol consumption, due to a "*lack of privacy in overcrowded ED*" (Physician, ED, Female); "*. . .hallway medicine where ED is overflowing is the norm for us*" (Physician, ED, Female). UEC settings were described as an "*ideal place for health promotion but overcrowding distracts staff from opportunity*" (Physician, ED, Female). There were some concerns about alcohol-related stigma, with patients (and some staff) feeling embarrassed to discuss alcohol consumption in a busy UEC setting, or patients being unable or unwilling to engage: "*Inebriated patients are unable to have authentic conversations. . .many alcoholics are unwilling to have honest conversations*" (Nurse, ED, Female). Alcohol-related stigma was expressed by a minority of respondents: "*They [alcoholic patients] somehow waste healthcare professionals' time instead of giving it to the more needed ones*" (Nurse, ED, Female). There were significant concerns raised about a lack of funding and resources for prevention in UEC, inadequate staff-facing, and patient-facing resources for advice-giving, and a lack of renumeration for prevention activities (such as UK Commissioning for Quality and Innovation (CQUIN) funding which supports improvements in the quality of services and the creation of new, improved patterns of care): "*It should not fall to UEC staff to pick up the pieces of longstanding underfunding of health, social, and psychiatric care without any additional time and support being made available*" (Physician, ED, Male); "*Well-meaning health promotion without funding for additional staff results in further pressure, further time per patient and ultimately delays to care for others*" (Physician, ED, Male).

Importantly, participants highlighted challenges with onwards referral and follow-up of patients with alcohol-related issues: "*screening is only useful if there is a meaningful intervention that follows identification of a problem*" (Physician, ED, Female); "*I'm happy to screen but I literally do not have the time to counsel and it seems like all services are a 2-year wait anyway*" (Nurse, ED, Non-binary). Key obstacles to referral included (a) '*few or inconsistent*' direct referral pathways (therefore a lack of standardisation in processes and a lack of clarity for staff regarding how, and where, to refer patients), (b) lengthy and complex referral processes (which increased workload for already over-burdened staff who chose to engage with SBIRT), and (c) long-waiting times for referred patients (meaning care was not timely–viewed to be a result of under-funding of services).

Respondents identified a range of facilitators of SBIRT. They specified that prevention activity required significant organisational commitment of time and resources, with visible support from leaders. Practical solutions were proposed, including raising awareness about the effectiveness of SBIRT, and the provision of workforce education and training on alcohol prevention: *"Training of staff and resourcing them to do so [the delivery of SBIRT]."* (Nurse, ED, Male). Specifically, participants desired training on how to approach patients in UECs to give them brief information, and how to signpost patients to appropriate support services: *"Knowledge of how to screen, access to advice and knowledge of how to refer."* (Nurse, ED, Female); *"online training tools for brief intervention"* (Physician, ED, Female); *"Training to increase confidence to start the conversation"* (Physician, ED, Female). This training was seen to be important since the UEC patient population was viewed to be particularly challenging with regards to addressing lifestyle behaviours. The capacity for authentic conversations was felt to be impacted by, for example, varying severity of mental or physical illness, inebriation (and level of consciousness), stigma (among both patients and healthcare professionals), and a perception that provision of brief advice would be *"information overload"* (Physician, ED, Male) for patients with comorbidities. Some proposed that education on SBIRT, and other health promotion interventions, should be introduced much earlier in the training pathway; that it should be embedded within pre-registration educational curricula with continuing professional development training for healthcare workers beyond registration.

The identified need for resources included adequate space, time, and materials and this was evident across UEC settings. Respondents referred to the need for private rooms to hold conversations with patients: *"appropriate areas for discussion i.e., not in corridor"* (Physician, ED, Male). They wanted capacity within their roles to undertake prevention or proposed that specialist staff were needed onsite to support UEC teams with prevention activities: *". . .improved support from specialist services especially OOH [out of hours]"* (Physician, ED, Male); *"Dedicated staff to do this."* (Physician, ED, Female); *"More time, space, and more trained staff."* (NP, Minor Injury Unit, Female); *"alcohol workers available to screen patients and provide the interventions and onward referrals"* (Physician, ED, Female). Respondents called for more evidence-based resources to be provided within UEC clinical areas, such as information leaflets and posters, which could be staff-facing (i.e., relating to SBIRT delivery) or patient-facing (i.e., relating to the availability of supportive services). Importantly, improvements in referral pathways were recommended by staff and this was a common theme across geographical regions: *"a clear, quick and easy to request referral pathway"* (Physician, Acute Medicine, Male). This could involve a review of health informatics, to produce for example, an *"easy to use flow-sheet set-up in operating systems"* (Nurse, ED, Female). Additional to improving referral processes, more service options were requested (including self-referral options), and adequate funding was advocated for follow-on services to reduce waiting times for patients.

**ii. Role conflict.**   Responses aligned with this theme came from a minority of individuals who responded to a survey item that health promotion was 'not' part of their role; they were primarily physicians and only one nurse. These individuals believed that health promotion should be undertaken by non-clinical staff, or staff in other healthcare environments such as primary care settings, or public health facilities: *"Train and recruit non-clinicians to undertake this role. It is unacceptable to expect medical and nursing staff to spend time and energy on health prevention in UEC given the current pressures on direct clinical care."* (Physician, ED, Male); *"These (health promotion interventions) are the GP's responsibility."* (Nurse, ED, Female). Other respondents, including physicians, nurses, and other health professionals, referred to a lack of clarity over who had responsibility for prevention within their role, with a *"lack of standardisation over who does what"* (ACP, Urgent Care Centre, Female); *"this is not seen as our role despite key motivators such as MECC frameworks meaning it should be"*

(Paramedic, ED Ambulance, Male). Some reported a "*lack of holistic culture in leadership*" (Physician, Urgent Care Centre, Female) and the existence of negative or judgemental attitudes towards alcohol prevention in the UEC workforce, which they believed were attitudes primarily held by physicians. These respondents attributed such attitudes to a general lack of interest in prevention (i.e., compared to diagnostics), a lack of empathy, a lack of understanding of the wider determinants of health, or the result of compassion fatigue in a time of pandemic and workforce crisis: "*[emergency medicine] has totally lost its compassion and ability to realise the importance of talking to patients*" (Physician, ED, Female). The view that health promotion was incompatible with the role of UEC personnel existed although it was a minority view, with most staff advocating prevention: "*There's still a bit of "it's not my job" but it's reducing"* (Nurse, ED, Male); "*we are all responsible for promoting health*" (Nurse, ED, Female); "*we should use any opportunity for prevention—it is not just for specific staff or roles*" (Nurse, ED, Female); "*It's important that we all use the opportunities we have to promote health. It only takes a short conversation*" (Paramedic, ED, Male). Individual variation in staff motivation for prevention in UEC was recognised: "*I think it comes down to the professional and their interest and willingness to make every contact count*" (ACP, ED, Female).

Key findings from the data, including enablers of SBIRT for alcohol prevention, are shown in Table 5.

## Discussion

The aim of the study was to ascertain the views and experiences of those working in urgent and emergency care (UEC) settings towards screening, brief intervention, and referral to treatment (SBIRT) for alcohol. Workers from diverse geographical regions, UEC settings and different occupational groups believe health promotion is part of their role and would be willing to deliver SBIRT to capitalise on 'teachable moments.' The willingness of the UEC workforce to engage in prevention is valuable given that prevention is globally advocated and forms a key component of World Health Organization policies [1], the Centers for Disease Control and Prevention 2022–2027 Strategic Plan [27], the EU Global Health Strategy [28] and the European Commission's EU4Health programme 2021–2027 [29]. In the UK, it is strongly advocated that prevention is embedded into all areas within healthcare services, and that healthcare staff use every contact with an individual to maintain or improve their health and wellbeing whatever their specialty or the purpose of the contact. The healthcare professions surveyed in this study play a key role in public health within and outside of UEC settings. Almost all our respondents held positive attitudes towards prevention in UEC, which has been identified elsewhere [30–32]. Our main area of focus was alcohol prevention, and most respondents believed that SBIRT for alcohol prevention is needed, appropriate, and acceptable to patients in UEC settings. In addition, many staff advocated that UEC settings would be suitable for promoting other areas of health (e.g., addiction–drugs, smoking, gambling; injury prevention; physical activity; diet and weight; violence; health screening; vaccination). Although, perceptions towards the level of difficulty in implementing these broader health promotion areas varied. Generally, workers who were based in ED, who were White, male, physicians and/or from the UK, were less likely to express positive attitudes towards promoting other areas of health compared to workers based in urgent care facilities, those from ethnic minority groups, females, nurses or ACPs and/or those based outside of the UK.

Although the concept of health promotion in UEC settings is gaining increasing attention [33], these clinical environments remain under-utilised for prevention [10] and as we have observed here, positive staff views towards health promotion in UEC (and specifically, alcohol prevention) do not always translate into clinical practice [34]. Alcohol-related disorders are a

**Table 5. Key views, practices, barriers, and enablers of SBIRT.**

| **Prevention is acceptable to the vast majority with recognised patient benefit** |
| --- |
| • Most UEC workers view health promotion to be part of their role, irrespective of geographical region or setting.<br>• Most are willing to deliver SBIRT and believe that SBIRT for alcohol is needed and appropriate across a range of UEC settings.<br>• Those with alcohol SBIRT experience believe it is acceptable to patients.<br>• Two-thirds agreed that UEC settings are suitable for promoting other health areas (e.g., drugs, smoking, injury prevention, physical activity, diet/weight, violence, health screening, vaccination).<br>• A minority believe health promotion is not part of UEC role, or they experience role conflict / uncertainty over who is responsible for prevention. |
| **Current practice of alcohol prevention in UEC settings is limited** |
| • Less than two-thirds have ever engaged in alcohol prevention, and almost half have never used an alcohol screening tool.<br>• Brief alcohol advice is more often practiced than alcohol screening or referrals.<br>• Knowledge, skills and confidence in SBIRT for alcohol vary with age, gender, ethnicity, occupation and geographical location. |
| **UEC remains a challenging environment to deliver alcohol prevention** |
| • Severe workforce shortages and the impact of the prolonged COVID-19 pandemic.<br>• The 3 most common barriers to SBIRT implementation are lack of time, lack of training, and lack of appropriate referral pathways.<br>• Other key challenges include alcohol-related stigma, lack of private physical space and lack of staff- and patient-facing resources.<br>• Despite increased recognition of the value of prevention, very few staff have had training relating to brief interventions for (any) lifestyle behaviours. |
| **Enablers of alcohol prevention as identified by the UEC workforce** |
| • Increased staff capacity in UEC (e.g., better staff-patient ratios, time for triage screening).<br>• Dedicated staff for health promotion / alcohol prevention in ED (e.g., health promotion advocate, social worker resident, alcohol liaison team: including out of hours).<br>• Guidelines and policies on alcohol (and other) screening, brief advice, and referrals.<br>• Access to private spaces for SBIRT conversations.<br>• Visible support for prevention activity from senior leadership.<br>• Increased funding for alcohol detox / abstinence / treatment programmes.<br>• Standardisation of roles to clarify who has responsibility for prevention.<br>• Workforce education and training (e.g., *topics*: importance / efficacy of SBIRT, reducing stigma, increasing knowledge and confidence in SBIRT; *delivered via*: online training tools; *delivered when*: pre- and post-registered healthcare education, UEC staff inductions).<br>• Availability of high-quality patient-facing resources (e.g., posters, leaflets)<br>• Quick and easy to access referral pathways providing timely and holistic patient support (i.e., standardisation of referral processes, and increased service options / availability).<br>• Mandating and/or rewarding SBIRT activity in UEC.<br>• Feedback (to staff / organisations) on screening and referral activity and outcomes. |

major avoidable burden to healthcare services, with a disproportionate impact on emergency attendances and hospital admissions. While figures greatly vary by country and setting (e.g., rural versus urban [35]), alcohol-related attendances can account for around 12–28% of ED attendances (UK: [36], [37]; Ireland: [38]; Australia and New Zealand: [39], USA: [40]). The prevalence of alcohol-related admissions may be an under-estimate as many alcohol-related ED attendances go undetected, due to failure to code alcohol-related issues e.g., if a patient presents with injuries [41]. Despite this, a large proportion of UEC workers has never engaged in alcohol prevention. However, in our sample, we observed marked differences between groups in that the youngest workers had less experience, knowledge, skills, and confidence for SBIRT, and female workers felt better supported for prevention within their organisations than male workers. We found differences between occupations, in that physicians were more likely to report having engaged in SBIRT, and having the knowledge, skills and confidence for SBIRT, compared to other professional groups. Although screening for alcohol problems is recommended in emergency settings [42], in our sample, those who had engaged with alcohol

prevention were more likely to have provided brief advice than screened for alcohol problems or made onward referrals. A US study identified that only one in six ED physicians consistently screened their patients for excessive drinking [43]. Further, the low number of staff making referrals to alcohol services widens the 'treatment gap' (patients who need treatment and/ or support, that do not receive it), which increases the risk of readmissions, perpetuating existing high readmission rates [44]. There is clearly a need to improve, simplify, and standardise the processes for alcohol screening and treatment referrals, to maximise the likelihood of patients receiving appropriate and timely support. Better health information technology (IT) systems are needed to improve adherence to screening and referrals. Although research on adherence to screening and treatment referrals processes in UEC is limited, there is scope to explore whether adherence could be facilitated by better health IT systems, with elements such as computerised clinician order entry applications—the process of providers entering and sending treatment instruction digitally or electronic reminders. Harnessing electronic health record systems and implementing 'single click' referrals systems are not universally implemented in UEC but could facilitate engagement with alcohol screening and treatment referrals and increase the accuracy of UEC records. This is clearly an area for further research and quality improvement initiatives in UEC.

There are many barriers to engaging with health promotion in UEC, but studies exploring the factors that help, or hinder prevention in UEC are sparse [11]. Our study is the first to identify that barriers to SBIRT in UEC are more likely to be reported by certain groups of workers (i.e., nurses, ethnic minorities, and UEC workers outside of the UK). This suggests that efforts to reduce barriers to SBIRT might benefit from targeted approaches.

Across our full sample, lack of time and workload pressures were the primary barrier to engaging in alcohol prevention. Over the last few decades, the demand on health services has risen dramatically, and this increase is expected to continue. The rising service demand has resulted in a rapid increase in workload in health and care settings, alongside chronic under-resourcing, and acute staffing shortages [45] that have been exacerbated across health and care services by the COVID-19 pandemic. Overcrowding in UEC settings is commonplace [46]. This impacts on availability of appropriate, private spaces for SBIRT delivery. Despite positive views towards SBIRT and the willingness of staff to implement it, significant investment of funds in UEC services is required to ensure an appropriate physical environment to deliver SBIRT and increase staff capacity to engage in such activities. This is increasingly evident given workforce challenges in the post-pandemic era. The allocation of dedicated staff based onsite to deliver SBIRT, such as ED-based drug and alcohol teams, 'health promotion advocates' or 'health champions' (e.g., [47]) with specific skills and expertise may be valuable. Visible support from senior leaders was flagged by our respondents as a potential enabler of prevention in UEC settings.

The second most reported barrier was lack of training, knowledge, and skills for SBIRT (i.e., related to both its effectiveness and the processes for delivery), which was highlighted by most respondents and aligns with previous findings [11]. There is a need to develop high-quality, evidence-based training for healthcare professionals that focuses on the provision of education about the wider determinants of health and the evidence for the effectiveness of SBIRT, coupled with practical guidance on SBIRT implementation. This could help to alleviate concerns (as we observed in our sample) about identifying appropriate patients for SBIRT, the use of alcohol screening tools, and how to start conversations about alcohol within the UEC setting, including strategies for managing resistance to advice. Patient stories might be valuable for communicating the importance of SBIRT for UEC patients, demonstrating positive outcomes, and helping to reduce alcohol-related stigma. This training could be delivered as continuing professional development across all healthcare professions, although further research

is required to explore whether this should be tailored to specific groups, such as occupation, level of seniority or level of UEC experience. As suggested by respondents in our study, there is a need to incorporate SBIRT training earlier in the education pathway into higher education curricula [48]. SBIRT has not historically been included as standard practice in undergraduate or postgraduate training for healthcare professionals either in the UK, or internationally, and so exposure to SBIRT during healthcare training is inconsistent. Yet, emerging studies have shown SBIRT training to be feasible with medical [49], nursing and advanced practice students [50].

Embedding SBIRT within the curricula across healthcare subject disciplines may help to normalise SBIRT practice in the future healthcare workforce and encourage healthcare professionals to utilise it as part of standard care. This may also encourage all healthcare professionals to see health promotion as part of their role, no matter their profession or place of work. Only a minority in our sample believed that health promotion is not part of their role; yet uncertainty relating to the professional role in alcohol misuse (and differences between professions) was found in another sample, albeit trainees [51]. Further, data from our free text responses suggests that lack of standardisation of job roles adds to this uncertainty. Broader workforce research shows that, in England for example, ACP roles lack standardisation and there is great variation in scope of practice, training and educational background of ACPs across the nation [52]. While comparable international studies may help to clarify the relevance of this in other geographical regions, this suggests that role standardisation may help to clarify who has responsibility for prevention in 'teachable moments' within, or outside of UEC settings.

We advocate for the development of online SBIRT training for staff since digital training is increasingly adopted in healthcare education worldwide [53] and online packages offer scalability, flexibility, portability, and adaptability. We have previously demonstrated the value of developing and implementing digital training packages for healthcare professionals (and healthcare trainees) that have had global reach (e.g., psychological wellbeing for health and care workers [54]).

While education and training are key, in order to maximise SBIRT practice, high-quality patient facing resources are needed at the point of care. Further research is needed into the type of resources that could be used by healthcare professionals as part of SBIRT implementation and would be well-received by patients. Early research explored the preferences of patients and visitors to the ED and found that despite innovations in the delivery of health education, more traditional approaches (e.g., books and brochures) were preferred to computer-based learning or classes, while video was the preferred modality for patients and visitors wanting to learn more about alcohol [55]. More than a decade on, there is value in exploring current patient information needs and their preferred modes of delivery for brief advice in UEC environments.

Although reported by the minority (8.8%, see Table 4), it is notable that some staff indicated that their willingness to engage in SBIRT for alcohol would be hindered by their own alcohol consumption; this was more commonly reported by workers from ethnic minority groups. Alcohol and substance misuse are not uncommon in the healthcare professions. A systematic review of 31 studies involving 51,680 participants in 17 countries, problematic alcohol use was reported by physicians and increased over time (from 16.3% in 2006–2010 to 26.8% in 2017–2020) [56]. A survey of ED doctors found that (despite being trained in the detection of alcohol misuse) 63% reported that they misuse alcohol at least once a month and 30% once or more a week [57]. The use of alcohol and illicit drugs can be associated with certain work situations and conditions (e.g., work-related stress, shift work, peer pressure, long working hours) and can lead to significant issues in the healthcare workplace, impairing work performance, increasing risk of poor decision-making, and increasing absenteeism, presenteeism and

inappropriate behaviour [58]. Problem alcohol consumption (i.e., binge-drinking, alcohol dependence) in doctors has been associated with occupational distress (e.g., psychiatric morbidity, burnout, job effort, work-life imbalance, coping with stress through self-blame or substances) [59]. Given the impact of the pandemic on occupational and psychological distress in healthcare workers [60] there may be a need for targeted intervention to reach healthcare workers at high-risk for alcohol/substance use prevention.

There was no attrition as all those who started the survey completed it. This study gathered views of personnel working in a range of UEC settings, across geographical regions (within and outside of the UK), varying in gender, age, and years of experience. The gender ratio of respondents (57.7% female) is not unexpected given that women comprise over 70% of the global health and care workforce and 77% of the UK NHS workforce, but the highest responding occupational group was physicians—a majority of whom are men. We had more responses from physicians and nurses than other professions, as would be expected since they are the largest occupational groups [45]. There was ethnic diversity in the sample (36.5% non-White) which broadly reflects that 25% of the UK NHS workforce are of Asian, black or another minority ethnicity [45], and most responses came from the UK, albeit 15.2% of our sample are from outside the UK, where the workforce may have a different demographic composition and skill mix.

## Study strengths and limitations

To our knowledge, this is the first study to describe the views, experiences, barriers, and enablers of SBIRT for alcohol prevention and make comparisons between UEC workers from different demographic groups, occupations, and UEC settings within, or outside, the UK.

Since UEC settings are under-researched and under-utilised for health promotion activities [9], our study makes a novel contribution to the limited evidence on barriers and enablers of SBIRT in UEC [11]. For context, this study took place during the COVID-19 pandemic which incurred significant economic cost to governments and society, impacting healthcare service delivery, globally. Our data collection occurred during a worldwide surge in both confirmed cases of the virus (March 2022: >468m; December 2022: >645m) and deaths (March 2022: >6m; December 2022: >6.6m) [61, 62], exacerbating an existing health workforce crisis [62], involving staffing shortages and a high prevalence of workforce distress, burnout, and trauma. This global public health emergency may have influenced staff capacity to take part in the study, and for responders, their views towards the viability of prevention activity in UEC.

As this was an open survey, we were unable to collect reasons for non-participation, although the pandemic is highly likely to have impacted on the survey response due to limited capacity of healthcare staff at the time of data collection. We chose not to use cookies or store IP addresses, for reasons of confidentiality and because staff working in UEC settings may have completed the survey on shared devices. Therefore, it is possible that some respondents may not have been unique visitors, although this is unlikely given the extreme pressures (i.e., COVID-19 impact, workforce shortages) being experienced by staff during the data collection period. The convenience sample approach may impact on external validity of the findings; some groups may be under- or over-represented. We have no data on non-responders and so cannot eliminate risk of response bias, although efforts to minimise bias are described in study procedures. The sample is not intended to be representative of all UEC settings globally but gives valuable and novel insights into the experiences and views of those in different regions and demonstrates that there are some differences in views and barriers to alcohol prevention between those working within the UK or elsewhere. Data gathered from free-text responses is likely to be constrained by the brevity of the response format and may not be as rich as data gathered using other qualitative methods, such as semi-structured interviews.

## Conclusions

Urgent and emergency care workers, from diverse geographical regions, settings, and occupational groups, are willing to engage in SBIRT, and it was broadly perceived to be appropriate and acceptable to staff and patients in UEC. However, there are challenges to implementation of SBIRT in UEC environments, including concerns about workload impacts in the context of sustained global healthcare workforce shortages and under-funding of services. UEC workers from diverse settings advocate for clear health promotion guidelines and policies, increased staff capacity and/or dedicated support teams onsite, SBIRT education and training, evidence-based patient-facing resources, appropriate physical spaces for SBIRT conversations and improved alcohol referral pathways to better funded services. This requires significant and sustained commitment of time and resources for prevention across healthcare organisations.

## Ethics approval and consent to participate

This study was performed in accordance with the Declaration of Helsinki and all relevant guidelines and regulations. The study researchers were trained in Good Clinical Practice. Information about the aims and conduct of the study were provided. Data were treated in confidence and analysed anonymously. Participants were informed that by voluntarily completing and submitting the online survey they were providing their written consent to take part. This survey study was the first part of a wider programme of research on alcohol prevention in urgent and emergency care, for which the protocol was approved by the University of Nottingham Faculty of Medicine and Health Sciences Research Ethics Committee on 19 January 2022 (FMHS 415–1121), with an extension to the study end date approved on 28 September 2022. The protocol was published on protocols.io (dx.doi.org/10.17504/protocols.io.bp2l6xkn1lqe/v1) on 30 June 2023 and the minimal dataset for this survey is available at The University of Nottingham Research Data Repository (doi: 10.17639/nott.7292).

## Supporting information

**S1 Table. STROBE checklist.**
(DOCX)

**S1 File. Example survey invitation.**
(PDF)

**S2 File. Questionnaire survey.**
(PDF)

## Acknowledgments

The authors thank the wider SCALES team, DREEAM Department of Research and Education at Nottingham University Hospitals NHS Trust, Andrew Tabner, Susan Dean, and Abejah Premakumar for assistance with survey promotion. We extend thanks to the patient and public involvement (PPI) group involved in the public consultation activity during the process of survey development.

## Author Contributions

**Conceptualization:** Holly Blake, Frank Coffey.

**Data curation:** Holly Blake, Mehmet Yildirim.

**Formal analysis:** Holly Blake, Mehmet Yildirim, Vinishaa Premakumar.

**Funding acquisition:** Holly Blake.

**Investigation:** Holly Blake, Mehmet Yildirim, Vinishaa Premakumar, Lucy Morris, Philip Miller, Frank Coffey.

**Methodology:** Holly Blake, Lucy Morris, Philip Miller, Frank Coffey.

**Project administration:** Mehmet Yildirim.

**Supervision:** Holly Blake.

**Writing – original draft:** Holly Blake.

**Writing – review & editing:** Mehmet Yildirim, Vinishaa Premakumar, Lucy Morris, Philip Miller, Frank Coffey.

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
