## [Decision Letter · Decision Letter 0]

29 May 2023

PONE-D-23-11156Attitudes and current practice in alcohol screening, brief intervention, and referral for treatment among staff working in urgent and emergency settings: a survey.PLOS ONE

Dear Dr. Blake,

Thank you for submitting your manuscript to PLOS ONE. After careful consideration, we feel that it has merit but does not fully meet PLOS ONE’s publication criteria as it currently stands. Therefore, we invite you to submit a revised version of the manuscript that addresses the points raised during the review process.

We look forward to receiving your revised manuscript.

Kind regards,

Joel Msafiri Francis, MD, MS, PhD

Academic Editor

PLOS ONE

Journal Requirements:

"The study was funded by Nottingham Hospitals Charity. The funders had no role in study design, data collection and analysis, decision to publish, or preparation of the manuscript.

We note that one or more of the authors is affiliated with the funding organization, indicating the funder may have had some role in the design, data collection, analysis or preparation of your manuscript for publication; in other words, the funder played an indirect role through the participation of the co-authors. If the funding organization did not play a role in the study design, data collection and analysis, decision to publish, or preparation of the manuscript and only provided financial support in the form of authors' salaries and/or research materials, please do the following:

(1) Review your statements relating to the author contributions, and ensure you have specifically and accurately indicated the role(s) that these authors had in your study. These amendments should be made in the online form.

(2) Confirm in your cover letter that you agree with the following statement, and we will change the online submission form on your behalf: 

Reviewers' comments:

Reviewer's Responses to Questions

**Comments to the Author**

1. Is the manuscript technically sound, and do the data support the conclusions?

Reviewer #1: Yes

Reviewer #2: Yes

2. Has the statistical analysis been performed appropriately and rigorously? 

Reviewer #1: Yes

Reviewer #2: Yes

3. Have the authors made all data underlying the findings in their manuscript fully available?

Reviewer #1: Yes

Reviewer #2: No

4. Is the manuscript presented in an intelligible fashion and written in standard English?

Reviewer #1: Yes

Reviewer #2: Yes

5. Review Comments to the Author

Reviewer #1: The research topic being addressed is of great public health importance however, there are limitations in generalisability and knowledge that is generated from these findings. You may consider making revisions on the objectives and analysis so as to bring forth novel insights in the topic.

Reviewer #2: I thank the authors for a well-structured and much needed evidence on the implementation of SBIRT in the urgent and emergency settings. The paper is well and extensively drafted, here are my comments to the authors.

General / Minor comments

Abstracts

Line 37-38: The sentence is incomplete, consider revising

Line 38-39: Authors indicated to involve “participants from any occupational group, working in urgent and emergency care (UEC) settings in any country or region”. Was this a multi-country study? I would advise indicating study setting in the abstract or title.

Line40 - 41: The sentence is incomplete, consider adding “we included 362 respondents (aged 21 - 65 years, 87.8% shift workers) from 7 occupational groups…”

Line 48 – 53: To keep the abstract succinct, I would advise the authors to have four sections in the abstract – Background (Including objectives), Methods, Results, Conclusion

Background

Please add the study aim in the last paragraph of the background section

Method

Line 103 – 109: The authors should consider moving the study objective to the last paragraph of the background

For clarity, I would advise the following flow for the method section

Study design and setting

Study population

Sampling and Sample size

Study procedures

Data Collection

Data Analysis

Ethical Considerations

Line 161 – 163: Did the Authors mean “Participant information sheet included the purpose of the study…”

Results

Table 1: Please summarise in the result section the variable “Target population” as documented for other variables’ findings

Table 1: The occupation group namely “linked profession” should be defined in the methods or table legend rather than defining it the results section

All tables should be referred in the text

P values indicated in the results should be added in the tables

Line 212 – 217: Authors should avoid repeating what has been presented in the table

Line 223: Consider removing the first statement

Line 242: The first statement should be removed

Line 259: The first statement should be removed

Line 315 – 316: Authors should ensure uniformity in presenting the results, all number should be accompanied by percentages

Line 318: change the subtitle “Analysis of qualitative free text responses” to Qualitative findings

Line 324: Change numbering from (ii) to (i)

Major comments

Line 151 – 154: It is not clear what link exists between COVID-19 pandemic and the rationale for the current study. Please elaborate.

Line 154 – 155: The authors mentioned the use of convenient sampling in this study. I suggest having a separate section on sampling and sample size. Was a minimum sample size determined? Authors should justify the sample size decided for this study.

Line: For how long was the link posted in the social media platforms?

Line 190 – 192: The authors mentioned the use of chi square in assessing the differences in views and practices relating to SBIRT according to demographic characteristics with statistical significance set at 5%. The findings depicting statistical significance in the differences in views should be indicated.

Line 347 -: Authors should consider developing a different theme on the strength of SBIRT. The results presented here do not align with the theme “The need to improve SBIRT delivery in UEC”

Limitation: Since the authors could not report on non-response, authors should acknowledge the possibility of response bias

6. PLOS authors have the option to publish the peer review history of their article (what does this mean?). If published, this will include your full peer review and any attached files.

Reviewer #1: No

Reviewer #2: **Yes: **Dr. Belinda J Njiro

---

## [Author Response · Author response to Decision Letter 0]

2 Jul 2023

PONE-D-23-11156

Attitudes and current practice in alcohol screening, brief intervention, and referral for treatment among staff working in urgent and emergency settings: a survey.

Dear PLOS ONE Editors,

Thank you for providing us with the opportunity to revise and resubmit this manuscript. We have endeavoured to address each comment below.

Response to Editorial team:

• We have updated the financial disclosure statement and provided this in the cover letter as requested. 

• We do not have any figures in the manuscript (only tables and supplementary files).

• The grant information provided in the ‘Funding Information’ and ‘Financial Disclosure’ sections should now match.

• The grant number has now been added into the paper, cover letter statement, and in the online form.

• Author contributions have been checked and are accurate. Please note that none of the authors are employed by Nottingham Hospitals Charity (the funder). The online system would not recognise this funder name. 

• The protocol is now published on protocols.io (dx.doi.org/10.17504/protocols.io.bp2l6xkn1lqe/v1) and the anonymized minimal dataset is available at The University of Nottingham Research Data Repository (doi: 10.17639/nott.7292). We have added a statement about this into the ‘ethics and consent to participation’ section, after the ethical approval reference. The data availability statement has been amended to include this information.

• Captions for Supporting Information files have been provided at the end of the manuscript and the in-text citations match accordingly.

Response to reviewers:

1. Is the manuscript technically sound, and do the data support the conclusions?

Reviewer #1: Yes

Reviewer #2: Yes

Thank you for this positive response.

2. Has the statistical analysis been performed appropriately and rigorously? 

Reviewer #1: Yes

Reviewer #2: Yes

Thank you for this positive response.

3. Have the authors made all data underlying the findings in their manuscript fully available?

Reviewer #1: Yes

Reviewer #2: No

The protocol is published on protocols.io (doi: awaiting assignment) and the minimal dataset is available at The University of Nottingham Research Data Repository (doi: 10.17639/nott.7292).

4. Is the manuscript presented in an intelligible fashion and written in standard English?

Reviewer #1: Yes

Reviewer #2: Yes

Thank you for this positive response.

Reviewer #1: The research topic being addressed is of great public health importance however, there are limitations in generalisability and knowledge that is generated from these findings. You may consider making revisions on the objectives and analysis so as to bring forth novel insights in the topic.

Thank you for this suggestion. We have conducted further analyses to explore whether there are differences in variables (attitudes and views) between demographic groups, setting (UK or international) and occupations. This has been added to the results and reflected on in the discussion.

Reviewer #2: I thank the authors for a well-structured and much needed evidence on the implementation of SBIRT in the urgent and emergency settings. The paper is well and extensively drafted, here are my comments to the authors.

Thank you for this positive feedback.

Abstracts

Line 37-38: The sentence is incomplete, consider revising.

The sentence was in abbreviated language for the abstract but has now been expanded.

Line 38-39: Authors indicated to involve “participants from any occupational group, working in urgent and emergency care (UEC) settings in any country or region”. Was this a multi-country study? I would advise indicating study setting in the abstract or title.

This was an international survey and we have amended the title and abstract to reflect this. We have chosen not to use the term ‘multi-country’ as we did not specify or target specific countries so this term could be mis-interpreted. 

Line40 - 41: The sentence is incomplete, consider adding “we included 362 respondents (aged 21 - 65 years, 87.8% shift workers) from 7 occupational groups…”

The sentence was in abbreviated form for the abstract but has now been expanded. We have chosen to use ‘There were…’ rather than ‘We included…’ to avoid any ambiguity.

Line 48 – 53: To keep the abstract succinct, I would advise the authors to have four sections in the abstract – Background (Including objectives), Methods, Results, Conclusion

Thank you for this suggestion, we have amended the abstract sub-headings.

Background

Please add the study aim in the last paragraph of the background section.

This has been actioned – aims and objectives have been moved to the end of the introduction.

Method

Line 103 – 109: The authors should consider moving the study objective to the last paragraph of the background.

This has been actioned – aims and objectives have been moved to the end of the introduction (and re-written for clarity).

For clarity, I would advise the following flow for the method section

Study design and setting

Study population

Sampling and Sample size

Study procedures

Data Collection

Data Analysis

Ethical Considerations

Thank you for the suggestion, this has been actioned.

Line 161 – 163: Did the Authors mean “Participant information sheet included the purpose of the study…”

Yes, the word ‘sheet’ has been added.

Results

Table 1: Please summarise in the result section the variable “Target population” as documented for other variables’ findings.

We have included the sub-heading ‘Study population’ (to align with sub-heading used previously in the methods).

Table 1: The occupation group namely “linked profession” should be defined in the methods or table legend rather than defining it the results section.

This is now included as a table legend as suggested:

aLinked profession: linked to / working directly with to ED patients, such as ED-based social prescribing; drug and alcohol liaison team; addiction psychiatrist; acute/critical care medicine physician.

All tables should be referred in the text.

All 5 tables are already referred to in the text.

P values indicated in the results should be added in the tables.

The only p values in the text related to a gender comparison which is not related to the content of the results table. Therefore, we prefer to leave this in the text.

Line 212 – 217: Authors should avoid repeating what has been presented in the table.

We have deleted this paragraph:

Physicians comprised almost half the study sample (n=176, 48.6%), with 146 (40%) from UK and 30 (8.3%) from overseas. Most respondents (n=318, 87.8%) reported working in shift patterns. Of shift workers, 74.1% (n=268) were from UK and 13.8% (n=50) were from overseas.

Line 223: Consider removing the first statement. Line 242: The first statement should be removed. Line 259: The first statement should be removed.

This statement has been removed from each section where it appeared.

Line 315 – 316: Authors should ensure uniformity in presenting the results, all number should be accompanied by percentages.

Thank you for the suggestion, we have included the missing percentages together with the denominator (those who responded to the training needs question item). 

Line 318: change the subtitle “Analysis of qualitative free text responses” to Qualitative findings.

This has been actioned.

Line 324: Change numbering from (ii) to (i)

This has been actioned.

Major comments

Line 151 – 154: It is not clear what link exists between COVID-19 pandemic and the rationale for the current study. Please elaborate.

This is important context indicating that the survey was undertaken during a time of extreme stress and pressure for the health and care workforce, with evidenced workforce shortages and burnout – this could potentially influence staff views on the practicalities of undertaking prevention activities in the UEC environment during this high-pressured time.

The wording has been tightened up to reflect this.

Line 154 – 155: The authors mentioned the use of convenient sampling in this study. I suggest having a separate section on sampling and sample size. Was a minimum sample size determined? Authors should justify the sample size decided for this study.

Thank you for this suggestion. This has been included in a separate section in the methods ‘Sampling and sample size’.

Line: For how long was the link posted in the social media platforms?

Data were collected from March to December 2022 and reminders were posted at least weekly via different distribution channels (e.g., emails, social media). This information is included in the manuscript.

Line 190 – 192: The authors mentioned the use of chi square in assessing the differences in views and practices relating to SBIRT according to demographic characteristics with statistical significance set at 5%. The findings depicting statistical significance in the differences in views should be indicated.

We have significantly revised the results to provide more information and also results of new analyses requested by reviewers.

Line 347 -: Authors should consider developing a different theme on the strength of SBIRT. The results presented here do not align with the theme “The need to improve SBIRT delivery in UEC”.

We have re-considered the theme labels and have re-labelled theme (i) as ‘Barriers and enablers of SBIRT delivery’ which directly aligns to section content.

Limitation: Since the authors could not report on non-response, authors should acknowledge the possibility of response bias

We have added this to study limitations while recognising that efforts were made to minimise bias as described in study procedures.

Not applicable. There are no figures in this paper.

REVIEWER 2

In the submitted manuscript titled: Attitudes and current practice in alcohol screening, brief intervention, and referral for treatment among staff working in urgent and emergency settings: a survey-title may need to be modified to reflect study objectives.

We feel the title reflects the study aims but have now modified the text to reflect the scope of the study, adding: ‘an international survey’.

I would like to commend the authors for the research efforts in addressing a topic of great public health importance. Your study explores the subject of (SBIRT) for alcohol among professionals working in urgent and emergency care (UEC) settings, which is undoubtedly of significant interest.

Thank you for this positive comment.

While I appreciate the insights provided by your descriptive findings, I must express some reservations regarding the external validity of the study design. The limitations in the design might make it challenging to generalize the findings to broader populations or settings. Consequently, the extent to which these findings contribute to the existing body of knowledge on the views and experiences related to SBIRT in UEC settings may be limited.

In light of this, I would like to suggest that you reconsider your research objectives and consider analyzing additional information that could enhance the value of your study. By incorporating additional variables or employing alternative methodologies, you may be able to provide novel insights and augment the existing understanding of this subject matter.

Thank you for your comment. Our study aims are objectives align with the proposal for which we received ethical approval, and the participants were provided with information that indicated our study focus. We therefore cannot add any additional variables or employ new methods.

However, to address concerns about the inclusion of only descriptive material (except for a small number of group comparisons that were included in the original manuscript), we have added some further analyses to compare attitudes and views according to demographic characteristics, setting (UK or international) and occupation. This has been added to the study objectives, and the findings are included in the results section and reflected on in the discussion. We believe this addresses your comment and adds value to the manuscript.

Once again, I commend you for your valuable contributions to the field and encourage you to refine and expand upon your work to further advance knowledge in this area. I believe that by addressing the aforementioned concerns, your research has the potential to make a substantial impact on the scientific community and inform future studies.

Thank you for this positive comment.

Below are additional points that need to be clarified

1. Title and abstract: 

i. It is important that the study design appears in the title

We had previously included the word ‘survey’ in the title, this has now been expanded to ‘an open, cross-sectional international survey’.

ii. Background seems to state the overall objective, you may consider modifying it to briefly discuss the topic under research.

Thank you for this suggestion. The aims and objectives have been modified for clarity and better alignment with survey data collection sections, and sub-headings in the results.

iii. The methodology appears somehow redundant, needs to capture more important details such as how data were analyzed, statistical tests performed etc, explain briefly that you collected qualitative and quantitative data 

We respectfully disagree that our methodology is redundant. To ensure the most important details were captured, we had carefully followed reporting guidelines in the “Strengthening the Reporting of Observational Studies in Epidemiology (STROBE) statement” and Checklist for Reporting Results of Internet E-Surveys (CHERRIES). 

In our original manuscript, we had already specified that both quantitative and qualitative data were collected: “The survey collected quantitative data including responses on a Likert scale and categorical responses with non-response options (i.e., ‘not applicable’, ‘prefer not to say’), and qualitative data in free text responses”.

Further, there was already a section called ‘Data analysis’ which reports how our qualitative data from free text responses was analysed, how quantitative data are described, and which tests were performed on quantitative data. 

However, we have now included more detail on the sampling strategy. We have also re-written the aims and objectives (as suggested elsewhere) which has made it clearer that we collected quantitative and qualitative data.

iv. Discussion should not be part of an abstract

Thank you for raising this, we have removed the word discussion and replaced it with conclusions.

2. Introduction: Background information and rationale to the problem under study has been clearly stipulated however the objectives/study aim has not been stated here but rather in the methods, I would recommend moving the study aim from the methods to the introduction (see the STROBE Checklist)

Thank you for this comment, we have moved the study aim to the introduction as suggested.

3. Methods

i. Study aim: should move to the introduction section.

This has been actioned.

ii. Study design: Adequate information on the study design has been provided.

Thank you for confirming.

iii. Questionnaire items: (line 133-142) you seem to have five objectives however the section mentioned in the tool seems not to capture all study objectives example, training needs to support future SBIRT practice.

The results directly map to study objectives and questionnaire items, although this was not clear enough. We have now improved the headings and heading levels to make this more obvious and added ‘training needs’ to section 5 of the survey items description. We believe this has improved the clarity.

iv. The study procedure has been comprehensively described; however, the authors do not explain the intended sample size prior to data collection that would also determine when the data collection should have been stopped.

We agree this was an important omission. This information has now been included in a separate section on sampling within the methods.

v. The ethical consideration has been explicitly stated.

Thank you for this positive comment.

4. Results

i. Table 1; I am not sure of the rationale for including age group 16-20 as it has not any participants.

We have deleted this row from the Table. Since this was a response option on the survey, we have added a Table footnote to ‘age’: aThere were no respondents in the 16-20 years category.

ii. Table 2; You could add a column that shows proportion out of the total (all cadres) to give an overall picture with regards to UEC.

Thank you for this suggestion, we have included a column showing the total and agree this helps with interpretation.

iii. P values have been presented, line 250, 251 and 262 however they do not seem to appear in the tables, kindly consider presenting them in the tables of results as well.

We have chosen to leave these in the text only, to avoid repetition, and because the gender comparison does not align with the content of the results table (Table 2).

iv. Line 324; “the need to improve SBIRT delivery in UEC” should be numbered (i)

This was a typographical error and has been corrected, thank you for picking this up.

v. Qualitative findings have been well described however, missing adequate primary data to back up the messages/themes being portrayed. Kindly consider adding more quotes for transparency.

Thank you for this suggestion. We have included many more quotes to illustrate the key messages and themes and believe this has improved the quality and flow of these sections.

5. Discussion

i. The discussion is too long; needs to be cut down significantly with a focus on findings that answer study objectives.

We have reviewed the discussion and respectfully prefer to retain the existing text as we cannot identify text that it would be helpful to remove. 

The reason for this, is that we have discussed all our findings systematically. We have ensured that we have discussed findings related to each study objective, in the same order that the objectives and results are presented in methods and results. These findings are discussed in the context of literature and/or practice with suggestions for the future as would be expected in a discussion (e.g., what it means, and where next). We do not believe there is any information included that is not relevant to the study objectives.

Please note that the reviewer has also requested that we undertake further analyses as part of this revision process, and so there are now significantly more results and therefore more findings to reflect on in the discussion as we did identify some very interesting group differences. However, we have attempted to keep any new text as short as possible (a few sentences).

6. Study limitations

i. I recommend that the subtitle reads strengths and limitations rather than limitations alone.

This has been actioned.

ii. This section is also too long, needs to be focused; less emphasis & text on COVID-19 pandemic (lines 550-557).

The section is focused on any survey reporting criteria from the CHERRIES checklist that could not be met, as study limitations. Therefore, the text needs to remain to ensure we adhere to the checklist. We included contextual impacts on survey response, risk of bias, limitations of open surveys (no data on non-responders) and lack of cookies/IP addresses to determine unique visitors, data representativeness and limitations of free-text responses as qualitative data.

The pandemic is very important context here in terms of its potential influence on survey response, and on healthcare professional’s views as to whether health promotion (prevention) activities are plausible in UEC settings, due to its impact on clinical care, workforce shortages and stress levels. We would like to ensure this context remains in the section, but we have re-written the section to make the relevance clearer and removed any redundant wording. We have also removed reference to the pandemic from the conclusion. 

iii. The authors should also consider providing additional study strengths and limitations. 

Thank you for this suggestion, we have now amended this section (and moved line 572 from conclusions into the study strengths). 

The section includes contextual impacts on survey response, risk of bias, limitations of open surveys (no data on non-responders) and lack of cookies/IP addresses to determine unique visitors, data representativeness and limitations of free-text responses as qualitative data.

7. Conclusion

i. Line 572 may not be suitable for the conclusion but rather the study strengths above.

Thank you for this suggestion, we have moved this from conclusion to study strengths and amended the subtitle to ‘Study strengths and limitations’.

---

## [Decision Letter · Decision Letter 1]

13 Aug 2023

PONE-D-23-11156R1Attitudes and current practice in alcohol screening, brief intervention, and referral for treatment among staff working in urgent and emergency settings: an open, cross-sectional international survey.PLOS ONE

Dear Dr. Blake,

Thank you for submitting your manuscript to PLOS ONE. After careful consideration, we feel that it has merit but does not fully meet PLOS ONE’s publication criteria as it currently stands. Therefore, we invite you to submit a revised version of the manuscript that addresses the points raised during the review process.

We look forward to receiving your revised manuscript.

Kind regards,

Joel Msafiri Francis, MD, MS, PhD

Academic Editor

PLOS ONE

Reviewers' comments:

Reviewer's Responses to Questions

**Comments to the Author**

1. If the authors have adequately addressed your comments raised in a previous round of review and you feel that this manuscript is now acceptable for publication, you may indicate that here to bypass the “Comments to the Author” section, enter your conflict of interest statement in the “Confidential to Editor” section, and submit your "Accept" recommendation.

Reviewer #1: All comments have been addressed

Reviewer #2: All comments have been addressed

Reviewer #3: (No Response)

2. Is the manuscript technically sound, and do the data support the conclusions?

Reviewer #1: Yes

Reviewer #2: Yes

Reviewer #3: Partly

3. Has the statistical analysis been performed appropriately and rigorously? 

Reviewer #1: Yes

Reviewer #2: Yes

Reviewer #3: Yes

4. Have the authors made all data underlying the findings in their manuscript fully available?

Reviewer #1: Yes

Reviewer #2: Yes

Reviewer #3: Yes

5. Is the manuscript presented in an intelligible fashion and written in standard English?

Reviewer #1: Yes

Reviewer #2: Yes

Reviewer #3: Yes

6. Review Comments to the Author

Reviewer #1: (No Response)

Reviewer #2: (No Response)

Reviewer #3: This is a descriptive study using a modest convenience sample. The paper in its present form is not ready for publication. (i) at >9000 words, it is far too long (ii) the findings are not generalizable and do not offer any new insights into the challenges of implementing SBIRT for alcohol concerns into ED and other urgent care settings, (iii), the study and findings are not adequately contextualised within the global literature and (iv) the authors did not take into account the previous reviewers comments about length. Given the reliance on descriptive statistics, limitatons to the design and the modest sample size, I would recommend reframing this paper as 1/2 brief reports.

7. PLOS authors have the option to publish the peer review history of their article (what does this mean?). If published, this will include your full peer review and any attached files.

Reviewer #1: No

Reviewer #2: No

Reviewer #3: No

---

## [Author Response · Author response to Decision Letter 1]

22 Aug 2023

Dear Editor,

Thank you for providing the opportunity to respond to the reviewer comments on the second submission. We have uploaded our third submission, in which we have endeavoured to respond to all comments either as a revision to the manuscript, or as a rebuttal.

Reviewer 1.

Thank you for confirming that: 

No further revisions are required. All comments have been addressed. The manuscript is technically sound, and the data support the conclusions. The statistical analysis has been performed appropriately and rigorously, and the data have been made available. The manuscript is presented in an intelligible fashion and written in standard English. 

Reviewer 2.

Thank you for confirming that: 

No further revisions are required. All comments have been addressed. The manuscript is technically sound, and the data support the conclusions. The statistical analysis has been performed appropriately and rigorously, and the data have been made available. The manuscript is presented in an intelligible fashion and written in standard English.

Reviewer 3.

Thank you for confirming that: 

The statistical analysis has been performed appropriately and rigorously, and the data have been made available. The manuscript is presented in an intelligible fashion and written in standard English. 

Our response to the second set of comments is:

This is a descriptive study using a modest convenience sample. 

This is accurate. As described in the methods, we adopted a convenience sampling strategy, targeting a sample size of 300-400 for a maximum sampling error of 5.0-5.8% [supported by reference 23]. 

With n=362 respondents, we therefore exceeded our minimum target sample size.

We are unable to change the study design but added further comparative analyses in response to the previous comments. In the discussion, we have reflected on the limitations of the sampling:

“The convenience sample approach may impact on external validity of the findings; some groups may be under- or over-represented. We have no data on non-responders and so cannot eliminate risk of response bias, although efforts to minimise bias are described in study procedures. The sample is not intended to be representative of all UEC settings globally but gives valuable and novel insights into the experiences and views of those in different regions and demonstrates that there are some differences in views and barriers to alcohol prevention between those working within the UK or elsewhere”.

Surveys with similar (or smaller) samples than ours have recently been published in PLOS ONE:

Schoultz M, McGrogan C, Beattie M, Macaden L, Carolan C, Dickens GL (2022) Uptake and effects of psychological first aid training for healthcare workers’ wellbeing in nursing homes: A UK national survey. PLoS ONE 17(11): e0277062. https://doi.org/10.1371/journal.pone.0277062

John-Henderson NA, Mueller CM (2020) The relationship between health mindsets and health protective behaviors: An exploratory investigation in a convenience sample of American Indian adults during the COVID-19 pandemic. PLoS ONE 15(11): e0242902. https://doi.org/10.1371/journal.pone.0242902

At >9000 words, it is far too long. The authors did not take into account the previous reviewers comments about length. I would recommend reframing this paper as 1/2 brief reports.

Thank you for your comment – we provide a rebuttal below.

The manuscript is not >9,000 words, the main text in our second submission was 8,145 words.

While this is significantly longer than our original submission, we were previously asked to add a large amount of new text (further detail required in every section), by three reviewers.

Reviewer 3 previously asked for more detail throughout the paper, and for all the results to be discussed in the context of the literature and in alignment with the study objectives. The reviewer also asked for further analyses in the results, these additional findings then also needed to be discussed. The reviewer requested previously that the qualitative section was expanded to include more quotes. We believe that we addressed all these requests for further detail, which naturally expanded the length – we tried to address requests as succinctly as possible and in the previous revision we removed some text from the limitations section. 

We have ensured that all our findings are discussed systematically and in relation to each study objective. Our findings are discussed in the context of literature and/or practice with suggestions for the future as would be expected in a discussion (e.g., what it means, and where next).

We have also reported our study in line with two checklists and to adhere to this, specific information and statements needed to be included (CHERRIES checklist, STROBE statement).

Reviewer 3 has now presented a new request for findings to be discussed in the context of global literature (i.e., requiring additional text and context) but at the same time, has asked for the paper to be converted to a short report.

We have considered this request but feel that it would not be possible to convert the paper to a short report that is high quality, while also (i) adhering to the reporting guidelines we used, (ii) addressing comments from three reviewers that all involved requests for further detail, and (iii) including the additional global context which is a new request from Reviewer 3. 

We do not believe it will add any value to split this paper into two. In our view, that would weaken the manuscript which currently benefits from the inclusion of both quantitative and qualitative data which is novel in this literature and findings have been triangulated in the interpretation and discussion.

It is important to note that two reviewers have already indicated that the additional text provided in the previous submission is satisfactory, and therefore removal of a significant amount of content at this stage would ‘undo’ the revisions made previously in response to all three reviewers.

Finally, the PLOS ONE guidelines clearly state: 

“Manuscripts can be any length. There are no restrictions on word count, number of figures, or amount of supporting information”.

The findings are not generalizable and do not offer any new insights into the challenges of implementing SBIRT for alcohol concerns into ED and other urgent care settings.

We respectfully disagree that the manuscript does not offer any new insights.

New insights are:

• Evidence synthesis shows there is very little data from the UK and none from England. Only one survey has been done before that included UK participants and that only included Scotland. Most of our respondents were from the UK, including England, Ireland, Scotland. Therefore, we add new data to the literature on challenges of implementing SBIRT for alcohol concerns into ED and other urgent care settings, from regions not previously explored.

• Evidence synthesis has called for more research in general to establish whether incorporating health promotion into the roles of staff in UEC settings is acceptable – therefore there is a view that our knowledge in this field is limited. We have responded to this call for more research in this field.

• Our study includes facilitators to health promotion in UEC settings which is not well captured in prior research (as evidenced in the literature). This information is needed to help us move forwards from barriers/challenges to enabling.

• We gathered views of personnel working in a diverse range of UEC settings, across geographical regions (within and outside of the UK), varying in gender, age, and years of experience. This has not been done previously.

• The study includes both quantitative and qualitative responses which strengthens and contextualises findings – prior studies have not done this.

• The global context has significantly changed in recent years (given the impacts of the COVID-19 pandemic) and this study presents current views, based on the current healthcare context. This is also relevant with the increasing burden on healthcare services and the increasing need (or pressures) for embedding prevention activity across a range of contexts and settings. We demonstrate that UEC workers are willing to engage in prevention but experience barriers to doing so, they suggest factors which facilitate prevention such as training, and our findings demonstrate the facilitators are largely not in place (e.g., few staff are being trained in SBIRT despite being willing to do it). This leads to recommendations. 

Regarding generalisability, we previously included this text in the limitations:

“The sample is not intended to be representative of all UEC settings globally but gives valuable and novel insights into the experiences and views of those in different regions and demonstrates that there are some differences in views and barriers to alcohol prevention between those working within the UK or elsewhere”.

The study and findings are not adequately contextualised within the global literature. 

Thank you for this valuable comment. 

We have already set our rationale in the context of global views towards health promotion and referred to research conducted across multiple countries in both the introduction and discussion, also using the terms ‘internationally’, ‘worldwide’, ‘global’ (etc) throughout our discussion.

However, we agree that more could be done to set the study findings in a global context with relation to health promotion policy, and ED attendance prevalence.

In the discussion, we have now added some context with relation to global policies (e.g., UK, Europe, US) and ensured that figures relating to alcohol-related attendances refer to variation across countries, with examples from UK, Europe, Australia and New Zealand, US. 

We have added a brief sentence to clarify where research is either unavailable in other geographical regions and/or may be needed.

We have added 7 new references to support the global context but have kept the additional text very succinct given this reviewer’s concerns about manuscript length.

---

## [Editor Report · Decision Letter 2]

23 Aug 2023

PONE-D-23-11156R2Attitudes and current practice in alcohol screening, brief intervention, and referral for treatment among staff working in urgent and emergency settings: an open cross-sectional international survey.PLOS ONE

Dear Dr. Blake,

Thank you for submitting your manuscript to PLOS ONE. After careful consideration, we feel that it has merit but does not fully meet PLOS ONE’s publication criteria as it currently stands. Therefore, we invite you to submit a revised version of the manuscript that addresses the points raised during the review process. Please submit your revised manuscript by Oct 07 2023 11:59PM. If you will need more time than this to complete your revisions, please reply to this message or contact the journal office at plosone@plos.org. Please include the following items when submitting your revised manuscript:A rebuttal letter that responds to each point raised by the academic editor and reviewer(s). You should upload this letter as a separate file labeled 'Response to Reviewers'.A marked-up copy of your manuscript that highlights changes made to the original version. You should upload this as a separate file labeled 'Revised Manuscript with Track Changes'.An unmarked version of your revised paper without tracked changes. You should upload this as a separate file labeled 'Manuscript'.If applicable, we recommend that you deposit your laboratory protocols in protocols.io to enhance the reproducibility of your results. Protocols.io assigns your protocol its own identifier (DOI) so that it can be cited independently in the future. For instructions see: https://journals.plos.org/plosone/s/submission-guidelines#loc-laboratory-protocols. Additionally, PLOS ONE offers an option for publishing peer-reviewed Lab Protocol articles, which describe protocols hosted on protocols.io. Read more information on sharing protocols at https://plos.org/protocols?utm_medium=editorial-email&utm_source=authorletters&utm_campaign=protocols.

We look forward to receiving your revised manuscript.

Kind regards,

Joel Msafiri Francis, MD, MS, PhD

Academic Editor

PLOS ONE

Journal Requirements:

**Additional Editor Comments:**

Thanks for sharing the revised draft. Thanks for your responses to the raised comments. I agree that the manuscript is extra long, which is acceptable for the PLOS journal. It would be helpful to reduce the number of references - 103. I suggest you reduced the references to maximum of 50. Also helpful to check all the references are in the correct format. Please proofread the whole document and be consistent in reporting (n,%) - check page 10, line 216. Be consistent with the reporting of p values - use 3 decimal points. Use one decimal point for proportions.

---

## [Author Response · Author response to Decision Letter 2]

28 Aug 2023

Dear Editor,

Thank you for the opportunity to complete minor revisions on our manuscript. 

Journal requirements: Please review your reference list to ensure that it is complete and correct. If you have cited papers that have been retracted, please include the rationale for doing so in the manuscript text, or remove these references and replace them with relevant current references. Any changes to the reference list should be mentioned in the rebuttal letter that accompanies your revised manuscript. If you need to cite a retracted article, indicate the article’s retracted status in the References list and also include a citation and full reference for the retraction notice.

We have reviewed the reference list and it is correct and aligns with author guidelines.

Additional Editor Comments: Thanks for sharing the revised draft. Thanks for your responses to the raised comments. I agree that the manuscript is extra-long, which is acceptable for the PLOS journal. It would be helpful to reduce the number of references - 103. I suggest you reduced the references to maximum of 50. Also helpful to check all the references are in the correct format. 

We have removed 40 references and adjusted the text accordingly. This has significantly reduced the number of references in this paper (and removed some of the text). 

Having reviewed the manuscript again, we believe that this is the maximum number we can remove without conducting a major re-write of the manuscript, and to ensure that we still (1) adhere to 2 sets of guidelines (CHERRIES and STROBE), and importantly, (2) retain all of the revisions made previously in response to 3 reviewers, over two rounds of review. 

We hope that this is acceptable to the Editorial team. 

We have checked the formatting of the references and ensured it is correct.

Please proofread the whole document and be consistent in reporting (n,%) - check page 10, line 216. Be consistent with the reporting of p values - use 3 decimal points. Use one decimal point for proportions.

Thank you for raising this. We have reviewed the whole manuscript carefully and ensured that the reporting is consistent.

---

## [Editor Report · Decision Letter 3]

1 Sep 2023

Attitudes and current practice in alcohol screening, brief intervention, and referral for treatment among staff working in urgent and emergency settings: an open cross-sectional international survey.

PONE-D-23-11156R3

Dear Dr. Blake,

We’re pleased to inform you that your manuscript has been judged scientifically suitable for publication and will be formally accepted for publication once it meets all outstanding technical requirements.

Kind regards,

Joel Msafiri Francis, MD, MS, PhD

Academic Editor

PLOS ONE
---

## [Editor Report · Acceptance letter]

18 Sep 2023

PONE-D-23-11156R3 

Attitudes and current practice in alcohol screening, brief intervention, and referral for treatment among staff working in urgent and emergency settings: an open, cross-sectional international survey. 

Dear Dr. Blake:

I'm pleased to inform you that your manuscript has been deemed suitable for publication in PLOS ONE. Congratulations! Your manuscript is now with our production department. 

Kind regards, 

on behalf of

Dr. Joel Msafiri Francis 

Academic Editor

PLOS ONE